

# Lepidopteran HMG-CoA reductase is a potential selective target for pest control

Yuan-mei Li[1], Zhen-peng Kai[1], Juan Huang[2] and Stephen S. Tobe[2]

[1] School of Chemical and Environmental Engineering, Shanghai Institute of Technology, Shanghai, China
[2] Department of Cell and Systems Biology, University of Toronto, Toronto, Ontario, Canada

## ABSTRACT

As a consequence of the negative impacts on the environment of some insecticides, discovery of eco-friendly insecticides and target has received global attention in recent years. Sequence alignment and structural comparison of the rate-limiting enzyme HMG-CoA reductase (HMGR) revealed differences between lepidopteran pests and other organisms, which suggested insect HMGR could be a selective insecticide target candidate. Inhibition of JH biosynthesis *in vitro* confirmed that HMGR inhibitors showed a potent lethal effect on the lepidopteran pest *Manduca sexta*, whereas there was little effect on JH biosynthesis in *Apis mellifera* and *Diploptera punctata*. The pest control application of these inhibitors demonstrated that they can be insecticide candidates with potent ovicidal activity, larvicidal activity and insect growth regulatory effects. The present study has validated that Lepidopteran HMGR can be a potent selective insecticide target, and the HMGR inhibitors (especially type II statins) could be selective insecticide candidates and lead compounds. Furthermore, we demonstrated that sequence alignment, homology modeling and structural comparison may be useful for determining potential enzymes or receptors which can be eco-friendly pesticide targets.

Corresponding author
Zhen-peng Kai, kaizp@sit.edu.cn

## INTRODUCTION

The traditional insecticides have made a major contribution to agriculture and health. However, as a result of improper use and the inherent shortcomings of some insecticides, many showed negative impacts on the ecological environment. Therefore, eco-friendly insecticides have received global attention in recent years. How to predict and avoid potential ecological risk in the initial phase of insecticide discovery is a problem that has not been fully resolved to date.

The insect juvenile hormones (JHs) are methyl esters of farnesoic acid 10, 11-epoxide (JH III) and related compounds, which function in the regulation of a number of insect physiological processes in insects including embryogenesis, larval and adult development, metamorphosis, reproduction, diapause, migration, polymorphism, and metabolism (*Nijhout, 1994*). JH biosynthesis proceeds in the corpora allata (CA) through the mevalonic acid (MVA) pathway, which insects share with most other organisms. By analogy with vertebrates, HMG-CoA reductase (HMGR) has been postulated to be a key enzyme

in the regulation of the MVA pathway in insects (*Feyereisen, Pratt & Hamnett, 1981*). JH biosynthesis in insect CA is inhibited *in vitro* by compactin (*Monger et al., 1982*); mevinolin (*Feyereisen & Farnsworth, 1987*; *Couillaud, 1991*); or fluvastatin (*Debernard, Rossignol & Couillaud, 1994*). However, compactin shows poor inhibition of JH biosynthesis *in vivo*. Only repeated injections into *Manduca sexta* larvae induced the black pigmentation characteristic of JH deficiency; the black pigmentation is always followed by death within approximately 24 h. In addition, compactin treatment by topical application has no effect on *M. sexta* larvae (*Monger et al., 1982*). Fluvastatin injected into locusts inhibited JH biosynthesis *in vivo*, but by 12 h, JH biosynthesis had almost fully recovered, with no discernible effects on either JH-regulated metamorphosis or oocyte maturation (*Debernard, Rossignol & Couillaud, 1994*). However, the use of HMGR inhibitors for pest control has not been fully explored. In addition, because HMGR is an enzyme which exists in most organisms, its status as an eco-friendly insecticide target remains unclear.

In the present study, we predict the possibility of HMGR as an eco-friendly insecticide target by using sequence alignment, homology modeling, and structural comparison. The effects of three commercial HMGR inhibitors on JH biosynthesis was assayed by using *M. sexta*, *Apis mellifera*, and *Diploptera punctata* as experimental animals *in vitro* to validate our predictions. Finally, the possible applicability of these compounds for pest control was demonstrated in this paper.

## MATERIALS AND METHODS

### Insects

Larvae of the tobacco hornworm, *M. sexta*, were raised from eggs provided by Carolina Biological Supply Company (Burlington, NC, USA) and reared on an artificial diet (Bio-Serv, NJ, USA) at 25 °C under a long-day (16 h light/8 h dark) photoperiod (*Bell & Joachim, 1976*). Pharate 5th instar larvae were set aside 4–7 h before lights off. The larvae molted within a few hours and were designated day 0. At the start of wandering, indicated by the appearance of a prominent dorsal vessel, the larvae were transferred to plastic vials containing vermiculite until pupation. Freshly ecdysed pupae were transferred to a chamber containing a tobacco plant and 10% sucrose under a long-day photoperiod into which the adult moths emerged (*Lee, Chamberlin & Horodyski, 2002*).

Newly emerged mated female *D. punctata* (day 0) were isolated from stock cultures. Mating was confirmed by the presence of a spermatophore. Stocks and isolated females were fed Lab Chow and water ad libitum, and were kept at 27 ± 1 °C and 50 ± 5% relative humidity with a 12 h light/12 h dark cycle (*Kai et al., 2009*).

Worker larvae of *A. mellifera* were collected from apiaries in Shanghai, China, and placed in an incubator at 34 °C and 80% relative humidity, fed a diet that was prepared with 40% pollen collected from combs and 60% honey. Fourth instar worker larvae were distinguished by the differences in maximum width of their head capsules (*Rachinsky, Tobe & Feldlaufer, 2000*).

## Chemicals

$L[^{14}C$-S-methyl] methionine was purchased from Amersham Biosciences (Piscataway, NJ, USA). HMGR inhibitors fluvastatin, pitavastatin and lovastatin and high-performance liquid chromatography (HPLC)-grade isooctane was purchased from Sigma-Aldrich (St. Louis, MO, USA).

## Bioassays

### Assays for JH biosynthesis assays in vitro

Rates of JH biosynthesis were determined *in vitro* by using the modified radiochemical assay (*Tobe & Clarke, 1985*; *Tobe & Pratt, 1974*). The radiochemical assays for JH biosynthesis were performed with CA from unfed day 1 fifth instars of *M. sexta*, day 7 adult female *D. punctata* and fourth instar workers of *A. mellifera*, respectively. HMGR inhibitors were dissolved in medium 199 (GIBCO) for assay as described previously (*Lee, Chamberlin & Horodyski, 2002*; *Kai et al., 2009*) and used on the same day that the inhibitors were prepared. Each pair of CA was incubated for 3 h at 30 °C in 100 µL of medium 199 with Hanks' salts, $L$-glutamine, 25 mM HEPES buffer (pH 7.2), 1.3 mM $Ca^{2+}$ and 2% Ficoll, containing $L[^{14}C$-S-methyl] methionine (40 µM, specific radioactivity 1.48–2.03 GBq/mmol) in the dark with gentle shaking. After incubation, both medium and CA were extracted with isooctane. The isooctane phase was removed and its radioactivity determined by liquid scintillation spectrometry. Inhibition of JH biosynthesis was calculated as percent activity compared with the control group (i.e., no HMGR inhibitor added). The $IC_{50}$ values for the test compounds were calculated by using GraphPad Prism version 5.0.

### Assays for JH biosynthesis in vivo

*Injection.* Injections of HMGR inhibitors (2 µL volume, and 1 µM concentration) in newly molted fifth instar *M. sexta* (day 0) were carried out using a 10 µL Hamilton-syringe. The final concentrations of the injected inhibitor in the hemolymph were approximately 4 nM. Control larvae were similarly injected, but with 2 µL of double distilled water. Larvae were first anesthetized by cooling on ice and then injected between the seventh and eighth spiracles near the horn, close to the posterior heart chamber. These animals were assayed for JH biosynthesis at day 1 using the method described in 'Assays for JH biosynthesis assays *in vitro*'. Each group of inhibitor-injected animals was compared with a group of water-injected animals treated concurrently.

*Topical application.* Solutions of HMGR inhibitors (5 µL) were applied to the dorsal abdomen of *M. sexta* fifth instars at day 0, and animals were assayed for JH biosynthesis at day 3 as described (see 'Assays for JH biosynthesis assays *in vitro*'). The concentration of the inhibitors (in 20% DMSO and 80% acetone) used in the bioassays was 100 µM. Each larva received 0.5 nmol inhibitor in the topical cuticular assays. Controls were treated with the solvent.

*Oral administration.* Newly molted fifth instars of *M. sexta* were immediately fed with 5 µL inhibitor solution (1 µM concentration); subsequently, these animals were fed on the

normal diet. Control larvae were similarly fed, but with 5 μL of double distilled water. JH biosynthesis in these treated animals was assayed one day 1ater by using the radiochemical assay.

### Assays for ovicidal activity on M. sexta

*M. sexta* eggs that had been deposited on a paper filter were briefly immersed in solutions of the HMGR inhibitor ($H_2O$ containing 0.2% DMSO as co-solvent, concentrations ranged from 1 μM to 1,000 μM). After the test solution had dried, eggs were maintained in Petri dishes. Five days later, the mortality (numbers of eggs that failed to hatch) was determined, relative to untreated controls (No eggs hatched after five days in either the treatment or control groups.).

### Assays for impact of feeding on M. sexta larval growth and mortality

Three groups of larvae were used for feeding assays. Newly hatched or newly molted *M. sexta* larvae were fed with HMGR inhibitor solution (2 μL for first and second instars, 3 μL for third instars, and 5 μL for fourth and fifth instars) at the beginning of the stadium, and then reared on the normal artificial diet until the next ecdysis. Larval mortality and insect growth were recorded after treatment.

### Statistics

Data presented as percentages were log-transformed before statistical analyses. Data were analysed by using a one-way analysis of variance (ANOVA) with Dunnett's multiple comparison test as the post hoc determination of significance by using GraphPad Prism version 5.0. Dose–response curves were prepared with GraphPad Prism. Values are expressed as mean $\pm$ standard errors (S.E.M.) with $N$ indicating the number of samples measured ($N$ is 8–20).

## Sequence alignment of HMGR

A sequence database of all known HMGR was collected from the literature and GenBank by using a combination of BLAST and keyword searches. Amino acid multiple sequence alignments for HMGR were constructed with ClustalW (*Thompson, Higgins & Gibson, 1994*) and adjusted by eye to ensure structural motifs were maintained. Poorly aligned regions and major gaps were deleted.

## Homology modeling

Because there was no crystal structure of insect HMGR, the homology models of HMGR of *M. sexta*, *A. mellifera*, and *D. punctata* were prepared respectively, to explore the three-dimensional structural differences of the HMGR from different organisms, especially the differences at their active site. A crystal structure of human HMGR in complex with Fluvastatin (PDB ID: 1HWI) was used as the 3D coordinate template for the homology modeling (*Istvan & Deisenhofer, 2001*). The homology models for HMGRs were generated by using the FUGUE and ORCHESTRAR modules in Sybyl. The initial model was optimized energetically by using the minimize program with steepest descent algorithm, AMBER7 FF99 as the force field and Gasteiger-Huckel as the atomic point charges. The minimization was terminated when the RMS gradient convergence criterion of

0.05 kcal/(mol Å) was reached. The qualities of these models were analyzed by PROCHECK (*Laskowski et al., 1993*).

## Docking calculations

A ligand lovastatin used for the docking studies with HMGRs from different organisms was constructed by using the 2D sketcher module in Sybyl. Minimum energy conformations of all structures were calculated with the Minimize module of Sybyl. The force field was MMFF94 with an 8 Å cutoff for nonbonded interactions, and the atomic point charges were also calculated with MMFF94 (*Halgren, 1999*). Minimizations were achieved with the steepest descent method for the first 100 steps, followed by the Broyden–Fletcher– Goldfarb–Shanno (BFGS) method until the Root-Mean-Square (RMS) of the gradient became less than 0.005 kcal/(mol Å) (*Head & Zerner, 1985*; *Kai et al., 2006*).

The Surflex-Dock (*Spitzer & Jain, 2012*) module implemented in the Sybyl program was used for the docking studies. The 3-D structures of *M. sexta*, *A. mellifera* and *D. punctata* were performed with homology modeling. Each inhibitor was docked into the binding site of the corresponding protein by an empirical scoring function and a patented search engine in Surflex-Dock applied with the automatic docking. Other parameters were established by default in the software.

## Molecular dynamics simulations

Docking calculations as described above were performed for each inhibitor in complex with HMGR in which energy was minimized and used the minimized program in Sybyl-X 2.0 with the optimization algorithm BFGS (*Head & Zerner, 1985*). The force field was AMBER7 FF99 and the atomic point charge was Gasteiger-Huckel for 500 steps to remove bad contacts (*Kai et al., 2006*). The system was equilibrated at 400 K for 0.1 ns followed by data collection, at regular intervals, for 10 ns. Each structure collected was subjected to 0.1 ns of simulated annealing to 300 K. The final 100 structures were energy-minimized and clustered using cut-off distance of <0.2 nm. AMBER7 FF99 was used for force field and Gasteiger-Huckel for charges in molecular dynamics simulation using the dynamics program of Sybyl.

## RESULTS

### Sequence analysis

As the aim of our study was to find an eco-friendly insecticide target, HMGR sequences from different species were aligned relative to that of *M. sexta*, and the identity values were recorded (Fig. 1). The HMGR sequences of Blattaria, Isoptera, Coleoptera, Hymenoptera, Homoptera and Diptera were significantly different from Lepidoptera, (Dunnett's multiple comparison test of identity values) (Fig. 1A). The identity values of more distantly related organisms, i.e., Malacostraca, Actinopterygii, Amphibia, Aves, Mammalia, Monocotyledoneae, and Dicotyledoneae were below 60% (Fig. 1B), in comparison with the identity value of the Lepidoptera (approximately 90%). This suggests that the HMGR sequences of Lepidopteran insects are significantly different from other organisms.

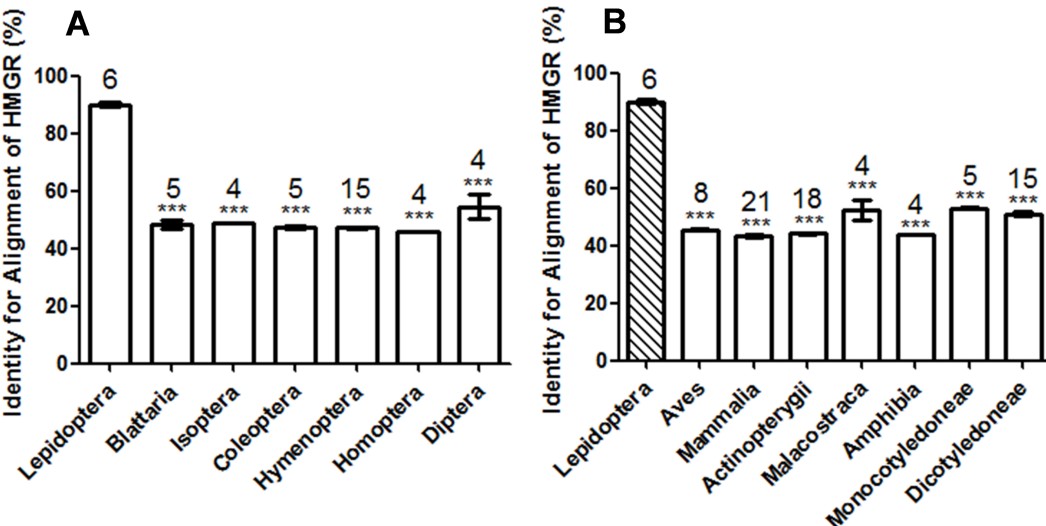

**Figure 1** **Comparison of the identity value of per species HMGR sequence relative to *Manduca sexta*.**
(A) Asterisks indicate significant differences between Lepidoptera and other orders of Insecta as determined by Dunnett's multiple comparison test following one-way ANOVA: ***, $P < 0.001$. (B) Asterisks indicate significant differences between Lepidoptera and other organisms other than Insecta as determined by Dunnett's multiple comparison test following one-way ANOVA: ***, $P < 0.001$.

## Homology modeling, docking and molecular dynamics optimization

With the goal of discovery of an eco-friendly insecticide target, the three-dimensional structures (especially the active site structures) of different species were analyzed. The HMGR of *M. sexta*, *A. mellifera* and *D. punctata* were selected for the structural comparison. The homology models of *M. sexta*, *A. mellifera* and *D. punctata* were generated using the crystal structure of *H. sapiens* (PDB ID: 1HWI) as the template. To select the best model, we checked the structural validity by PROCHECK (http://services.mbi.ucla.edu/SAVES). The geometry of the final refined models were evaluated with Ramachandran plot calculations computed using the PROCHECK program. The torsion angles of $\varphi$ and $\psi$ (the two torsion angles of the polypeptide chain, also called Ramachandran angles, describe the rotations of the polypeptide backbone around the bonds between N-C$\alpha$ called $\varphi$ and C$\alpha$-C called $\psi$.) in the generated model was represented in the Ramachandran plot as shown in Fig. S1. The Ramachandran plot showed 89.7% of the residues of *M. sexta*, 88.2% residues of *A. mellifera* and 90.5% residues of *D. punctata* existed in the most favored regions. The percentages of residues in disallowed regions of *M. sexta*, *A. mellifera* and *D. punctata* are 0.0%, 0.1% and 0.3%, respectively. This indicated that the backbone dihedral angles, phi and psi, of the three homology models were reasonably accurate.

One HMGR structure was assembled with the same four subunits. Each two adjacent subunits constituted a ligand binding pocket, which means one HMGR contained the same four binding pockets. Lovastatin, a commercial HMGR inhibitor, was used to identify the binding pocket of the aforementioned HMGR structures with docking calculations. Molecular dynamics (MD) simulations of the three homology models complexed with lovastatin as the ligand were performed for 10 ns to obtain the stable and low energy

conformations. By reporting the root mean square deviation (RMSD) of the protein structure from the starting model, the receptor changes in structure and reaches a relatively stable conformational minimum after approximately 3 ns. The conformations with the lowest energy of the final 100 structures from the MD simulation were selected as the final structures.

## Structure comparison

Both subunits that constituted the binding pocket of *A. mellifera* and *D. punctata* formed hydrogen bonds with lovastatin, whereas only one subunit of *M. sexta* can form hydrogen bonds with the ligand (Arg 579 and Lys 680 of chain A). This suggested that the binding pocket of *M. sexta* is more flexible.

The surface properties of the binding pocket of the above three structures were defined using MOLCAD calculations (an interactive visualization of molecular scenarios) in Sybyl to analyse these binding pockets. Figure 2 sketches the molecular surfaces of the pockets of these four HMGRs. The cavity of *M. sexta* was smooth and did not penetrate deep into the structure compared with the cavities of *A. mellifera* and *D. punctata* (Fig. 2). This result was in accordance with the results of hydrogen bond interaction. The electrostatic potential of the binding pockets of *M. sexta* was positively-charged, because the whole surface of the pocket was colored in yellowish green, whereas that of *A. mellifera* was electroneutral (Fig. 2). The front of the binding pocket of *D. punctata* was colored with blue, which suggests that some part of this pocket is electronegative. The lipophilic potential of the binding pocket of *M. sexta* was lipophilic, colored with brown, whereas that pocket of *D. punctata* was more hydrophilic, colored with blue. The green color suggested that the pocket of *A. mellifera* is neutral (Fig. 2). These results show that the active pockets of the three species are different, suggesting that it is possible to design eco-friendly insecticides using differences in surface properties. Thus, the insect HMGR may represent a potential eco-friendly insecticide target.

The surface properties of these three pockets suggests that increasing the molecular volume, electronegativity and lipophilicity of the ligand can strengthen the binding affinity between ligand and HMGR of *M. sexta*, whereas it also can weaken the binding affinities with other species.

## Effects of HMGR inhibitors on JH biosynthesis *in vitro* and *in vivo*

To validate the results of sequence alignment and structure comparison, three commercial human HMGR inhibitors (statins) were used for the assay of JH biosynthesis *in vitro* and *in vivo* in *M. sexta*, *A. mellifera* and *D. punctata*. Lovastatin is a type I statin. Fluvastatin and pitavastatin are type II statins. The IC$_{50}$ value of each compound is shown in Table 1. For the lepidopteran pest *M. sexta*, all the compounds have potent inhibitory activity on JH biosynthesis. However, these compounds have little or no effect on *A. mellifera*, which suggests that HMGR inhibitors tested in the present work are safe for honeybees. Similarly, the inhibitory effects on *D. punctata* were much lower than on *M. sexta*. The above results suggest that insect HMGR (in particular, lepidopteran HMGR) might be an eco-friendly insecticide target.

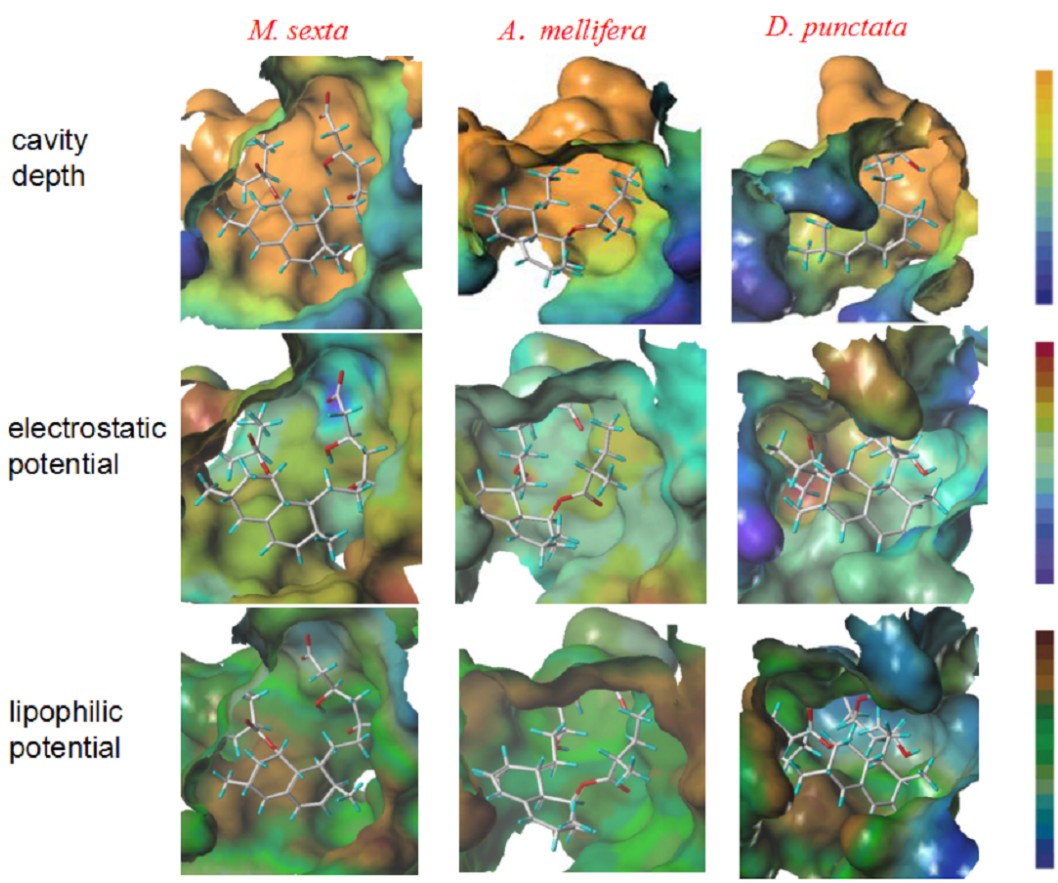

**Figure 2** **Structure comparison of binding pockets of *M. sexta*, *A. mellifera* and *D. punctata* HMGR.** The molecule in the pocket is lovastatin. In the presentation of cavity depth, the deep blue colour represents the outermost surface of the structure, whereas the orange colour represents the deepest part of the cavity. In the presentation of molecular electrostatic potential, the deep blue colour represents the most negative potential, whereas the deep red colour represents the most positive potential. In the presentation of the molecular lipophilic potential, the deep blue colour represents the most hydrophilic parts of the surface, whereas the deep brown colour represents the most lipophilic parts of the surface.

Type II statins have a greater effect than type I for the inhibition of JH biosynthesis in *M. sexta*. The $IC_{50}$ value of lovastatin was 99.4 nM, which is much higher than that of fluvastatin or pitavastatin (their $IC_{50}$ values are 5.1 nM and 5.2 nM, respectively). These results suggest that type II statins should be good lead compounds for new insecticide design. In addition to their effects *in vitro*, the statins also showed significant effects on JH production by *M. sexta* following treatment *in vivo*.

### Injection

Following injection of the statin into newly molted fifth larval instar *M. sexta*, JH biosynthesis was assayed after 3 h with significant inhibitory effects apparent. After 3 h, the inhibition of fluvastatin, pitavastatin and lovastatin was $64.1 \pm 5.2\%$, $61.4 \pm 5.8\%$ and $60.6 \pm 5.8\%$, respectively.

**Table 1    The IC$_{50}$ values of HMGR inhibition of JH biosynthesis *in vitro*.**

| Compound | Structure | *M. sexta*, IC$_{50}$ value (nM) | *A. mellifera*, IC$_{50}$ value (nM) | *D. punctata*, IC$_{50}$ value (nM) |
|---|---|---|---|---|
| Fluvastatin | | 5.11 | 18100 | 150.0 |
| Lovastatin | | 99.45 | No effect | 884.7 |
| Pitavastatin | | 5.23 | 157500 | 395.2 |

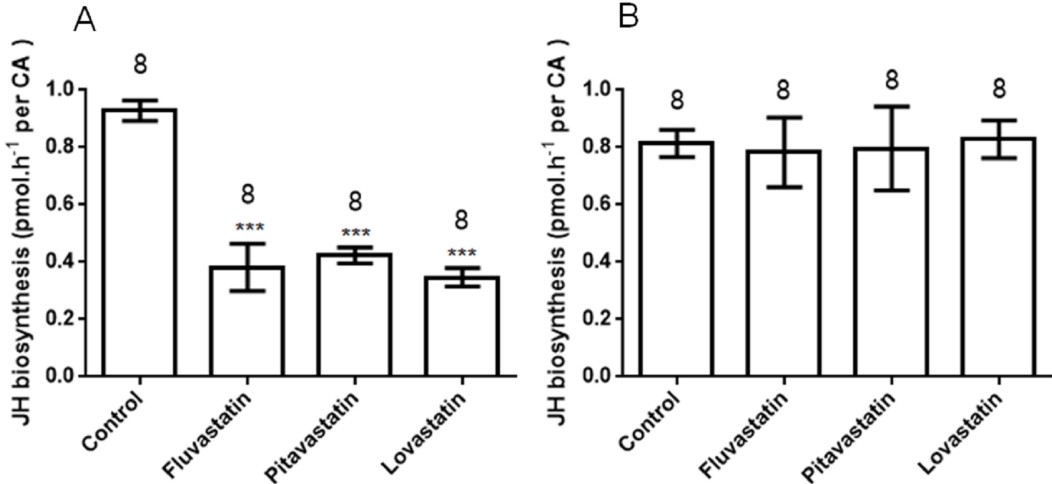

**Figure 3    JH biosynthesis following oral treatment of various inhibitors (statins) (A) and topical cuticular application of the same inhibitors (B).** Each bar represents the mean ± SEM. Asterisks indicate significant differences between inhibitor- and water-fed groups of animals as determined by Dunnett's multiple comparison test following one-way ANOVA: ***, $P < 0.001$.

### Effects on JH biosynthesis following oral administration

In addition to the effects in the injection bioassays, the statins also showed a significant effect on JH production following oral administration (Fig. 3A). In bioassays at 1 μM, inhibition of JH biosynthesis by fluvastatin, pitavastatin and lovastatin was 58.9 ± 8.9%, 54.4 ± 3.1% and 62.6 ± 3.4%, respectively. This suggests that HMGR inhibitors can also inhibit JH biosynthesis following oral administration.

**Table 2  Ovicidal effect of HMGR inhibitors on *M. sexta* eggs at different concentrations.**

| Compound | Concentration ($\mu$M) | Number of eggs | Mortality (%) |
|---|---|---|---|
| Fluvastatin | 1,000 | 60 | 100 |
| | 100 | 110 | 76.1 |
| | 10 | 86 | 11.9 |
| | 1 | 107 | 9.5 |
| Pitavastatin | 2,000 | 69 | 100 |
| | 200 | 90 | 100 |
| | 20 | 90 | 40.3 |
| | 2 | 70 | 38.3 |
| Lovastatin | 1,000 | 66 | 100 |
| | 100 | 99 | 99.99 |
| | 10 | 96 | 30.9 |
| | 1 | 88 | 37.8 |
| Control | 0 | 100 | 0 |

*Topical application*

In topical cuticular assays, no compound demonstrated any effect on JH biosynthesis (Fig. 3B); this might be attributable to the poor cuticular penetration of the reagents, and indicates that HMGR inhibitors are unlikely to be contact insecticides.

## Pest control application
*Ovicidal effects*

The three compounds also demonstrated significant activity on viability of *M. sexta* eggs (Table 2). At a concentration (100 $\mu$M), the mortality of eggs following treatment with fluvastatin, pitavastatin and lovastatin was 76.1%, 100% and 100%, respectively. A concentration of 50 $\mu$M of these compounds gave about 50% inhibition.

*Larvicidal effects following oral administration of statins*

The experiment in 'Effects on JH biosynthesis following oral administration' demonstrated that the statins have a significant effect on JH biosynthesis following oral administration. Accordingly, a stomach toxicity test was performed. We first determined which instars of *M. sexta* were most sensitive to the statins following feeding at high concentration (1,000 $\mu$M) in 1st, 2nd and 3rd instars, respectively, and recorded the mortality in Table 3. All larvae that commenced feeding from the 1st stadium died prior to pupation, and most died in the 1st stadium. Larvae fed from the 2nd stadium were also sensitive to the statins with a high mortality (above 85%). However, when the treatment commenced from the 3rd stadium, the mortality before the next molt was less than 10%, and mortality just prior to pupation was less than 40%. For all the inhibitors, the earlier instars were more sensitive than the later instars.

We then topically treated larvae with different concentrations of the statins, commencing with 1st instars, and recorded larval mortality (Table 4). The statins showed significant larvicidal activity at 100 $\mu$M. The IC$_{50}$ values of fluvastatin, pitavastatin and lovastatin were 2,101 $\mu$M, 63.0 $\mu$M and 298.3 $\mu$M, respectively.

**Table 3** Mortality following feeding with statins at 1,000 μM during the first three stadia.

| Compound | Feeding treatment | Mortality before next molt (%) | Mortality before pupation (%) |
|---|---|---|---|
| Fluvastatin | From 1st instar | 100 | 100 |
| | From 2nd instar | 51 | 92 |
| | From 3rd instar | 6 | 13 |
| Pitavastatin | From 1st instar | 82 | 100 |
| | From 2nd instar | 80 | 85 |
| | From 3rd instar | 10 | 40 |
| Lovastatin | From 1st instar | 100 | 100 |
| | From 2nd instar | 90 | 100 |
| | From 3rd instar | 20 | 90 |

**Table 4** Larval mortality following treatment of 1st instars.

| Compound | Concentration (μM) | Mortality before 3rd instar (%) | Larval mortality (%) |
|---|---|---|---|
| Fluvastatin | 1,000 | 100 | 100 |
| | 100 | 25 | 25 |
| | 10 | 8.3 | 8.3 |
| | 1 | 16.7 | 16.7 |
| Pitavastatin | 1,000 | 100 | 100 |
| | 100 | 100 | 100 |
| | 10 | 41.7 | 41.7 |
| | 1 | 25 | 41.7 |
| Lovastatin | 1,000 | 100 | 100 |
| | 100 | 33.3 | 33.3 |
| | 10 | 0 | 8.3 |
| | 1 | 0 | 0 |

In the dead larvae, the most striking characteristic was the darkening of the cuticle in some animals as well as molting disturbances (Fig. 4). This is consistent with the phenomenon of the inhibition of JH biosynthesis (*Monger et al., 1982*).

### Growth regulation

A long-term feeding study with low concentrations of inhibitors was performed to identify their effects on growth. We fed larvae with inhibitors starting with 1st instars, and recorded the number of days from hatching to the 5th stadium (larvae which died before the 5th stadium were not recorded). For fluvastatin, the number of days from newly hatched larvae to the 5th stadium at 100 μM, 10 μM and 1 μM was 17.3, 13.6 and 13.5 days, respectively. In the control group (not fed inhibitors), the interval was 11.9 days. It appears that the HMGR inhibitors significantly slowed the growth rate of *M. sexta*. Figure 5A shows the difference in growth more clearly. Three larval groups hatched on the same day; one was fed normal food as control, one was treated with 100 μM of fluvastatin, and the other was treated with 1 μM of fluvastatin. The difference in size was readily apparent by comparison with the control 4th instars. The other statins also showed the same growth effect as fluvastatin (Fig. S2).

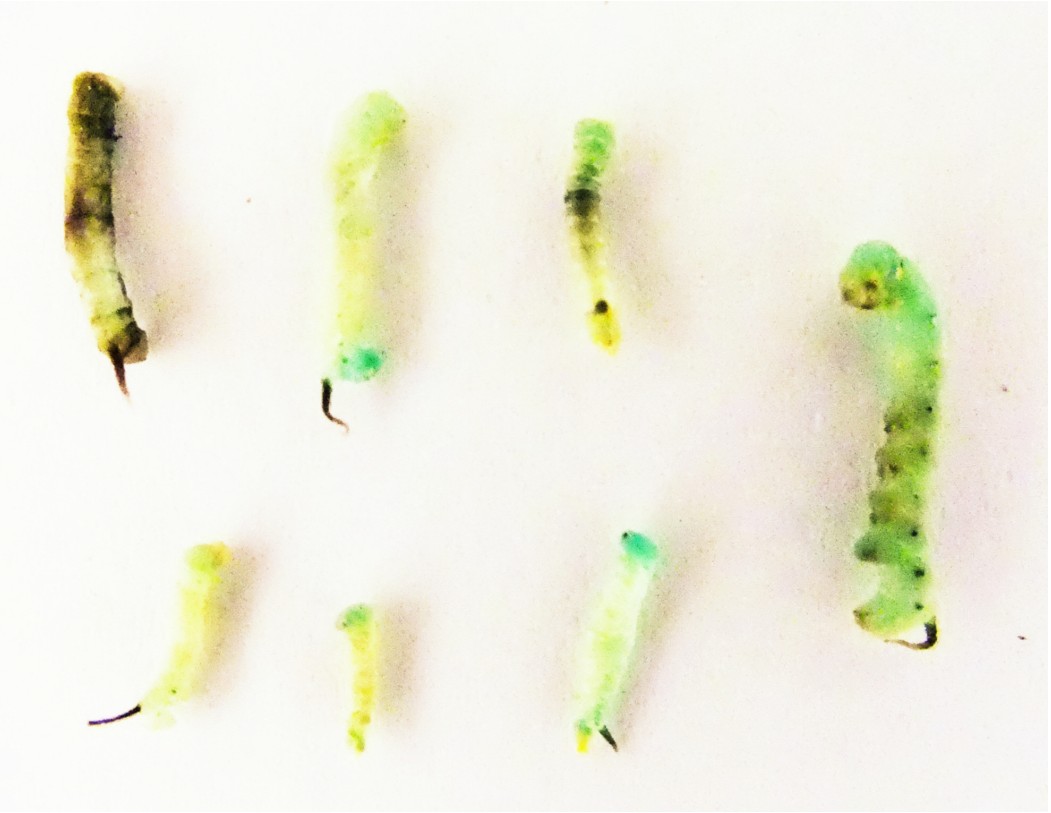

**Figure 4** **Dead larvae following feeding with 1,000 µM of fluvastatin as 1st instars.** Twenty animals were used in this treatment.

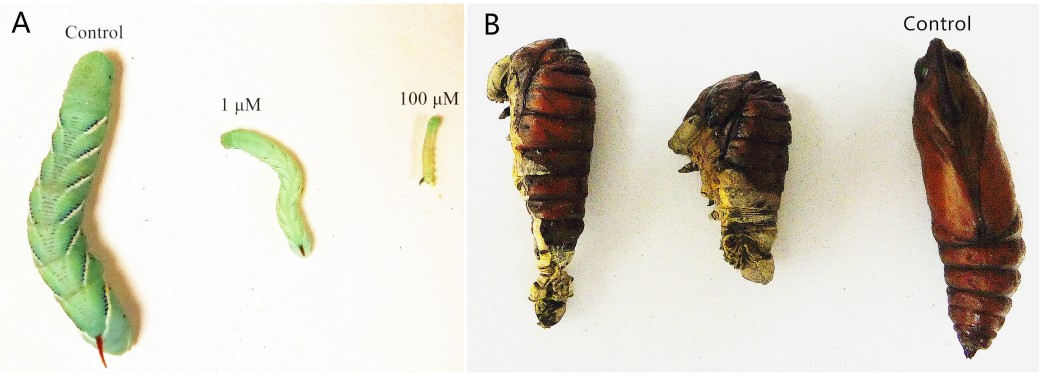

**Figure 5** **Developmental arrest and growth retardation in *M. sexta* following fluvastatin treatment.** There were twenty animals for each group. (A) Three larval groups hatched on the same day; one was fed normal food as the control; the others were treated with 1 µM and 100 µM of fluvastatin, respectively. The difference in size is readily apparent in the control 4th instars (10 days after feeding). (B) Newly hatched larvae were treated with 1 µM fluvastatin, most died in the process of pupation as a result of malformation.

Following treatment with a low concentration (1μM) of the statins, we observed that pupation of these larvae was not normal. Most larvae died in the process of pupation as a result of malformation (Fig. 5B). This suggests that the statins can be potent insect growth regulators.

## DISCUSSION

Currently, there is an on-going need for the discovery and development of new insecticides to combat growing problems associated with resistance, environmental pollution, accumulation of pesticide residues in the food chain and detrimental effects on non-target organisms. Hence, the need for eco-friendly insecticides with safe and novel modes of action or targets is becoming increasingly important. In this study, we have not only focused on elucidating new eco-friendly insecticide targets and lead compounds, but also attempted to provide an empirical method for eco-friendly insecticide discovery.

Insect JHs are a group of structurally related sesquiterpenoids that regulate a number of physiological processes including embryogenesis, larval and adult development, metamorphosis, reproduction, pheromone biosynthesis, diapause, migration, polymorphism, and metabolism (*Nijhout, 1994*; *Kerkut & Gilbert, 1985*; *Gilbert, Granger & Roe, 2000*). To our knowledge, the occurrence of JHs and related sesquiterpenoids such as methyl farnesoate is confined to animals in the Arthropoda. It has been demonstrated that the design of JH mimics or anti-JH agents is an effective strategy for insecticide discovery. Screening new targets involved in JH biosynthesis has been a subject of study for two decades (*Bede et al., 2001*). As HMGR has been postulated to be a key enzyme in the regulation of the MVA pathway in insects, some HMGR inhibitors (statins) have been used to investigate their effects on JH biosynthesis. Compactin, mevinolin and fluvastatin have been demonstrated to be potent inhibitors of JH biosynthesis *in vitro*, whereas studies of their effects *in vivo* are incomplete (*Monger et al., 1982*; *Couillaud, 1991*). Thus, to date, HMGR has not been used as a potential insecticide target. Furthermore, whether HMGR inhibitors have detrimental effects on non-target organisms remains unknown.

We predicted and evaluated the ecological safety of HMGR by using sequence alignment and structural comparison. Sequence analysis showed that the Lepidoptera differ from other organisms. Zapata et al. tested the effects of two HMGR inhibitors, fluvastatin and compactin, on HMGR activity of *Blattella germanica*. Both compounds significantly inhibited the enzymatic activity at a high concentration (50 μg per animal) by approximately 25% *in vivo* (*Zapata et al., 2002*). The inhibition by fluvastatin on *M. sexta* HMGR was approximately 60% at a low concentration (0.8 ng per animal in an injection assay and 2 ng per animal following oral administration, respectively) in our study. BLAST showed that the identity value between *B. germanica* HMGR and *M. sexta* HMGR was 47%. Bacterial HMGR has a low sequence identity value compared with *M. sexta* HMGR. Lovastatin inhibited *Pseudomonas mevalonii* at a high concentration ($K_i$ value = 0.53 mM) (*Hedl & Rodwell, 2004*). However, the inhibitory effect of lovastatin on *M. sexta* was much greater than on *P. mevalonii* ($IC_{50}$ value 99 nM). The identity value between *P. mevalonii* HMGR and *M. sexta* HMGR was 23%. In our experiment, the identity value of *A. mellifera* and

*D. punctata* versus *M. sexta* was 48% and 50%. Three HMGR inhibitors have no or little effect on *A. mellifera* and *D. punctata*; however these compounds are potent inhibitors in *M. sexta*. This suggests that there might be a link between the sequence alignment data and inhibition.

The sequence and 3-D structure (in particular, the molecular potential surface properties) of lepidopteran HMGR differs from other organisms (*A. mellifera* and *D. punctata*) in this study. Assays of JH biosynthesis in the presence of HMGR inhibitors in different insect species showed that those inhibitors have potent effects on the lepidopteran pest *M. sexta*, but are much less effective on *A. mellifera* (Hymenoptera) and *D. punctata* (Blattodea). This confirms our suggestion regarding the value and assessment of the ecological safety of HMGR as an insecticide target candidate.

The applicability to pest control is crucial in the evaluation of the potential of a chemical to act as an insecticide. Previous studies indicated that HMGR was the control point in JH biosynthesis in *M. sexta* (*Monger et al., 1982*). As a consequence of limited experimental data on the application of HMGR inhibitors for pest control, we tested the effects *in vivo* of three HMGR inhibitors on *M. sexta*. Our present study revealed that HMGR inhibitors can be potential insecticide candidates with excellent ovicidal activity, larvicidal activity and growth regulatory effects. In the fat body, HMGR was crucial to vitellogenesis and reproduction. Short-term assays showed that HMGR inhibitors reduce the protein levels and enzymatic activity of HMGR, and long-term experiments revealed that fluvastatin impairs embryo development (*Zapata et al., 2002*). Our work clearly indicates that HMGR is a key enzyme in embryogenesis, larval and adult development and metamorphosis. In *Agrotis ipsilon*, fluvastatin also disrupted normal spermatophore transfer (*Duportets et al., 1998*). It suggested that insect HMGR can be an insecticide target and its inhibitors could be insecticide lead compounds.

We conclude that an empirical method of discovery of eco-friendly insecticides encompasses the prediction of ecological safety of insecticide target candidates and the probability of the application for pest control. The steps for ecological safety prediction are as follows:

(1) Collect sequence data of insecticide target candidates from all species of interest.
(2) Perform sequence alignment of each species and compare to the selected target pest. Statistically analyze the identity values from sequence alignments. If there is no difference between pest and non-target organisms, this candidate is a likely to be an eco-toxic insecticide target. If not, go to the next step.
(3) Perform structural comparisons and docking studies with ligands of pests and other non-target organisms. If there is no difference between their structures (especially the binding pockets) or binding affinities, this candidate is a possible eco-toxic insecticide target. If not, it could be an eco-friendly insecticide target. New eco-friendly insecticides can be designed based on structural differences.

Although this method of prediction cannot replace the requisite toxicity tests, it can avoid unnecessary waste, save manpower, material and time in the discovery of new eco-friendly insecticides.

## CONCLUSION

We have demonstrated that insect HMGR can be a potential selective insecticide target, and its inhibitors can be potential selective insecticides. Our research should be helpful for designing new selective insecticides. Furthermore, we have demonstrated that sequence alignment, homology modeling and structural comparison can be used to determine which enzymes or receptors could be selective pesticide targets. Pest control applications have shown that the HMGR inhibitors are potential insect growth regulators, especially for lepidopteran pest control.

## ACKNOWLEDGEMENTS

The authors thank Dr. Frank M. Horodyski (Ohio University, Athens, OH) for providing bioassay equipment.

### Funding

This work was supported by a grant from the National Natural Science Foundation of China (No. 21402122). It was also supported by Natural Science Foundation of Shanghai (No. 14ZR1440600), and the Scientific Research Foundation for the Returned Overseas Chinese Scholars, Ministry of Education of PRC (No. ZX2015-9). The funders had no role in study design, data collection and analysis, decision to publish, or preparation of the manuscript.

### Grant Disclosures

The following grant information was disclosed by the authors:
National Natural Science Foundation of China: 21402122.
Natural Science Foundation of Shanghai: 14ZR1440600.
Scientific Research Foundation for the Returned Overseas Chinese Scholars, Ministry of Education of PRC: ZX2015-9.

### Competing Interests

Stephen S. Tobe is an Academic Editor for PeerJ.

### Author Contributions

- Yuan-mei Li performed the experiments, analyzed the data, prepared figures and/or tables.
- Zhen-peng Kai conceived and designed the experiments, performed the experiments, analyzed the data, contributed reagents/materials/analysis tools, wrote the paper, prepared figures and/or tables.
- Juan Huang performed the experiments, analyzed the data.
- Stephen S. Tobe contributed reagents/materials/analysis tools, wrote the paper, reviewed drafts of the paper.

## Data Availability

The raw data on bioactivities are included in Fig. 3 and in Tables 1–4.

## Supplemental Information

Supplemental information for this article can be found online at http://dx.doi.org/10.7717/peerj.2881#supplemental-information.

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
