# Peer review of "Lepidopteran HMG-CoA reductase is a potential selective target for pest control"

_PeerJ, doi:10.7717/peerj.2881_

## Round 0.1 · original submission · Minor Revisions

All four reviewers ranged from positive to very positive about this MS, and I agree with those assessments. The reviewers provided some excellent guidance for improving this MS further (please note that Reviewer #3 included a marked-up PDF with a number of important suggestions regarding wording, etc.). Please make and/or respond to the reviewers' requested revisions as per the directions provided by PeerJ (including a detailed rebuttal letter). Thank you for submitting your work to PeerJ.

·

Basic reporting

This manuscript is very well written and easy to understand. A few comments on various areas:

- Figures 1 and 3 show nicely the biological variation in the testing. The authors should somehow show the reproducibility of the findings in Figures 4 and 5. E.g., if each group had 10 animals in it, show pictures of all ten animals. On that note, what use is Fig. 4 if there is no controls?
- some of the figure legends/labelling seem incomplete. E.g., Fig. 5B - are we to assume that the left two pupae treated?
- Discussion - one very obvious consideration seems to be overlooking in the authors' excitement about their findings - how useful are inhibitors that may also target humans going to be? Neonicotinoids, e.g., are very useful because their effects in humans are seen at orders of magnitude higher concentrations than in insects. I appreciate that the authors are only using inhibitors designed for human targets here as proof-of-concept but if they seen an effect there, it is reasonable to assume that there may also be effects in the other direction, once an inhibitor is optimized for Lepidopterans. This needs to be addressed

Experimental design

The obvious experiment that is missing here is a comparison of the effects of one or more of the tested inhibitors' effects in M. sexta versus one of the other species (A. mellifera or D. punctata). Again, I appreciate that none of the compounds has been optimized for Lepidopterans but this comparison seems so obvious and would add a lot to the story.

Validity of the findings

The data are generally sound, although, as noted above, it would help to show the biological variability in the experiments behind Fig. 4 and Fig. 5.

Reviewer 2 ·

Basic reporting

The manuscript is well written and is clear and unambiguous.

Experimental design

The experimental design is well done.

Validity of the findings

I see no problems with the validity of the findings. The modeling appears well done (though modeling is outside my expertise) and the JH biosynthesis assays and effect on insects appears well done.

Additional comments

The work is well done and the statins clearly effect JH biosynthesis. Because of the massive use of statins by humans, I see very little likelihood that they will be used for insect control. Because of their polarity, the fact that they have very little effect when applied topically is what I would expect.

Reviewer 3 ·

Basic reporting

No Comments

Experimental design

No Comments

Validity of the findings

No Comments

Additional comments

General Comments:

This is a fairly well written research report that may have over-reached a bit from a conceptual standpoint. This type of lab/computer based analysis does not really supplant a long term ecological field study to establish the environmental impact of a pesticide treatment. The authors should acknowledge that in the Discussion.

There are many editorial issues with the manuscript that could be addressed. See the attached edited version of the MS.

The overuse of the terms “potent” and “eco-friendly” make this paper in some places seem more like an advertisement than a research report.

One item that the authors might consider in a revision is that the preponderance of their cited references are pretty old (25-35 yrs). A good example of this is on l. 344 where to support a statement of studies of the effects of the statins on JH biosynthesis in vivo are incomplete, they reference a paper from 25 yrs ago and another from 34 yrs ago. It seems incredible that no work in this area has been done since 1991?

Title and Abstract: I’m wondering if the terms selective or selectivity might be more appropriate than “eco-friendly.” Eco-friendly and potent are two terms that seem to have been overused in the manuscript. Also, in the Abstract the authors might include some of the key results that they have for the orthopteriod, Diploptera punctata. This would illustrate that their approach is relevant to both the Holometabola and Hemimetabola. The Abstract does not include any information on the potential impact of these treatments on mammals and birds, which would be other major components of the ecosystem where the treatments would take place. It also does not include a description of the empirical method for insecticide discovery that the authors refer to in the Discussion. Perhaps a brief description of this should be included in the Abstract.

References: The style of the journal names seems to be pretty variable (some abbreviated, some not abbreviated, etc.). The authors should work on this with the editorial staff. Latin names of some of the taxa in the titles in the references were not always italicized.

Figures: I have made suggested revisions for most of the figure legends directly on the MS.
Figure 1B. This is problematic because the data for the Lepidoptera were not included in the figure and it is not clear if a statistical test was carried out for this histogram in panel 1B. Also, P-values less than 0.001 are usually just noted as P<0.001. It would be helpful to see the sample sizes for each mean reported somewhere on the figure or in the legend.
Figure 1. Comparison of the identity value of per species HMGR sequence relative to Manduca sexta. (A) Asterisks indicate significant differences between Lepidoptera and other orders of Insecta as determined by Dunnett’s multiple comparison test following one-way ANOVA: ****, P<0.001. (B) The identity values of organisms other than Insecta. There were no significant differences.


Figure 3:
Figure 3. JH biosynthesis following oral treatment {of what?}(A) and topical cuticular application (B) of various inhibitors (statins). Each bar represents the mean ± SEM (N = 8). Asterisks indicate significant differences between inhibitor- and water-fed (control) groups of animals as determined by Dunnett’s multiple comparison test following one-way ANOVA: ***, P<0.001. {I would report the dose/concentration of the various treatments in the figure legend as well as the sample size N. You might also add the word “Oral” above the histogram in A and “Topical” above the histogram in B}

Annotated reviews are not available for download in order to protect the identity of reviewers who chose to remain anonymous.

Reviewer 4 ·

Basic reporting

I believe that the results are of interests for the community and should be published. However, the use of the word “discovery” in the title is quite strong, considering that the impact of HMGR inhibitors on juvenile hormone biosynthesis has been previously investigated (Monger et al., 1982, for example).

Experimental design

In general, the experimental design is clearly written. A few minor changes, additions, modifications would be suggested as follows:
Pg 11, line 156-157: Clarification on the structural motifs maintained: identification of those motifs in a multiple sequence alignment in a figure would be helpful to the reader.
Pg 12, line 170: The description refers to “all ligands” but only one ligand is actually discussed in the results section.
Pg 12, Section 2.6: Was there a validation of the docking procedure? A simple self-dock of Lovastatin in its human structure would be sufficient to determine the reliability of the docking.
Pg 14, line 198-202: The number of sequences used for each Order would help understand the depth of the analysis.
Pg 15, line 224-225: It was never mentioned if the binding site used for docking and computational analysis is the same as the one typical for statins in HMGR
Pg 15, line 227-230: What is the r.m.s.d. value reached? An r.m.s.d. time series figure would validate the reached conclusion.
Pg 15, line 233: This information deserves further explanation. The residues involved in those hydrogen bonds should be named. A figure representing these interactions for each model should also be shown; examples of such figures are well presented in the 2001 Science paper cited for the human HMGR structure used as template (PDB: 1HWI).

Validity of the findings

Validity of the findings is reasonable.

---

## Round 0.2 · Minor Revisions

Thank you for your response. Could you please include a DOCX file that shows *all* of the changes that you have made in the MS? Currently the DOCX file supplied does not show changes made as per suggestions of Reviewer #3. It seems from looking through that the changes were made, but it is very hard for me, as an editor, to pinpoint each change.

Also, I'm not 100% why the title was changed. If this is the new title, however, then it currently contains a typo. Should be:

"Lepidopteran HMG-CoA reductase is a potential selective target for pest control"

---

## Round 0.3 · accepted · Accept

The authors have responded adequately to the various comments from the reviewers. Thanks to both the reviewers and the authors for making this a smooth process and for working to improve the MS.